# Clinical *versus* Ultrasound Measurements of Hyomental Distance Ratio for the Prediction of Difficult Airway in Patients with and without Morbid Obesity

**DOI:** 10.3390/diagnostics10030140

**Published:** 2020-03-03

**Authors:** Cristina Petrișor, Sebastian Trancă, Robert Szabo, Robert Simon, Adrian Prie, Constantin Bodolea

**Affiliations:** 1Department of Anaesthesia and Intensive Care II, “Iuliu Hațieganu” University of Medicine and Pharmacy Cluj-Napoca, Clinical Emergency County Hospital of Cluj, 400006 Clinicilor 3-5, Cluj-Napoca, Romania; sebi_tranca@yahoo.com (S.T.); robsz2011@yahoo.com (R.S.); simon.robert08@gmail.com (R.S.); adrian_prie@yahoo.com (A.P.); cbodolea@gmail.com (C.B.); 2Anaesthesia and Intensive Care 1, Clinical Emergency County Hospital Cluj-Napoca, 400006 Cluj-Napoca, Romania; 3Anaesthesia and Intensive Care Department, Municipal Hospital Cluj-Napoca, 400139 Cluj-Napoca, Romania

**Keywords:** hyomental distance, difficult airway, ultrasound

## Abstract

Purpose: To describe the correlation between clinically measured hyomental distance ratio (HMDR_clin_) and the ultrasound measurement (HMDR_echo_) in patients with and without morbid obesity and to compare their diagnostic accuracy for difficult airway prediction. Methods: HMDR_clin_ and HMDR_echo_ were recorded the day before surgery in 160 consecutive consenting patients. Laryngoscopy was performed by a skilled anesthesiologist, with grades III and IV Cormack–Lehane being considered difficult views of the glottis. Linear regression was used to assess the correlation between HMDR_clin_ and HDMR_echo_ and receiver operating curve analysis was used to compare the performance of the two for predicting difficult airway. Results: The linear correlation between HMDR_clin_ and HDMR_echo_ in patients without morbid obesity had a Pearson coefficient of 0.494, while for patients with morbid obesity this was 0.14. A slightly higher area under the curve for HMDR_echo_ was oberved: 0.64 (5%CI 0.56–0.71) versus 0.52 (95%CI, 0.44–0.60) (*p* = 0.34). Conclusion: The association between HMDR_clin_ and HDMR_echo_ is moderate in patients without morbid obesity, but negligible in morbidly obese patients. These might be explained by difficulties in palpating anatomical structures of the airway.

## 1. Introduction

Preoperative difficult airway prediction is one of the most important elements for the preanesthetic evaluation and comprises several clinical signs. Recognizing patients with difficult airways will alert the physician and will allow all safety measures to be taken before induction [1]. As none of the clinical signs has absolute diagnostic value and none can exclude difficult intubation, there is a continuous search for a predictive test with improved diagnostic accuracy that identifies patients at risk for airway problems, such as the inability to intubate and/or ventilate. Imagistic techniques like computed tomography, magnetic resonance imaging, radiology and ultrasonography have been investigated in the attempt of identifying parameters with optimal sensitivity and specificity for difficult airway anticipation. More than fifteen ultrasonographic measured parameters have been used in prospective studies, among which are the measurements of the hyomental distance (HMD) and hyomental distances ratio (HMDR) [2]. 

HMDR measured clinically (HMDR_clin_) was investigated for difficult airway prediction in anesthesia more than 10 years ago. It was reported that a cutoff value of 1.2 discriminated between patients with difficult airways versus easy laryngoscopy [3]. The diagnostic accuracy of this parameter varied widely in subsequent studies, suggesting that evaluation of clinical parameters might be operator-dependent or that there might be difficulties regarding the palpation of the hyoid bone in certain patients [4,5,6,7,8]. In the last years, ultrasound measurements were applied for the hyomental distance ratio (HMDR_echo_), also with variable sensitivity and specificity [9,10,11,12,13]. However, until now, each of the previous studies on HMDR focused on a single technique of measurement, either clinical or ultrasonographical, and no study compared the two.

We hypothesized that ultrasound measurements might be more precise compared to the clinical measurements. This might be especially important for patients in whom the hyoid bone is not easy to palpate, like the morbidly obese patients. These are believed to have higher difficult intubation incidences [14,15]. There are no comparative studies on HMDR measured clinically versus sonographically for patients with and without morbid obesity.

The aims of our study are to assess and describe the correlation between HMDR_clin_ and HMDR_echo_ in patients with and without morbid obesity and to compare their diagnostic accuracy for the prediction of difficult airways.

## 2. Materials and Methods

This prospective observational study was conducted in a university surgical hospital, after the Ethical Committee approval was obtained (Project No. 30303, 6 November 2018). All adult patients signed the informed consent form prior to being included. The study was conducted from 1 January 2019 until 1 July 2019. All consenting consecutive adult patients scheduled to undergo elective surgery under general anesthesia with oro-tracheal intubation were prospectively included. Patients with neck deformities, laryngeal neoplasm, or a history of radiotherapy, cervical spine surgery, or emergency surgery requiring rapid sequence inductions were excluded. 

During the standard preanesthetic evaluation, the patient’s body mass index was calculated as weight divided by the height squared (BMI). Standard clinical signs of difficult intubation were recorded by the attending anesthesiologist and included the Mallampati score, upper lip bite tests as previously defined, neck circumference, mouth opening, and head extension [16,17]. The attending anesthesiologist also recorded the clinical measurements for HMD in neutral and maximal hyperextended positions [3]. The measurements were performed from the hyoid bone as palpated and the tip of the mandible (the chin). HMDR_clin_ was obtained from the ratio of the hyomental distance measured with the head placed in maximal extension and the hyomental distance with the head placed in neutral position, as previously described [3]. The anesthesiologist was blinded with regard to the ultrasound measurements and was not a member of the study team [16,17].

Members of the study team performed the ultrasound measurements with a standard curvilinear probe placed in the mid-saggital plane in the submandibular region [18,19]. HMDR_echo_ was obtained from the ratio of the hyomental distance with the head placed in maximal hyperextended position and the hyomental distance measured with the head in neutral position [18,19]. The hyomental distance was measured between the anterior border of the hyoid bone and the posterior aspect of the symphisis menti (Figure 1). The ultrasound measurements were performed the day before surgery and anesthesia with a curvilinear ultrasound transducer (Venue 50, General Electrics, Boston, Massachusetts, USA).

Before the induction of general anesthesia with orotracheal intubation, patients without obesity were positioned supine, with the head in sniffing position, while the obese patients were placed supine in a ramped position with the sternal notch and the external auditory meatus on a horizontal line, these being the current standard positions for direct laryngoscopy [20]. After preoxygenation, induction was performed by the attending anesthesiologist with fentanyl, propofol, and neuromuscular blocking agents adjusted per body weight in all patients. Laryngoscopy was performed with a standard Macintosh curved blade and the Cormack–Lehane view of the glottis visualization was recorded. Grades III and IV were considered difficult view during laryngoscopy [21]. Patients considered to have difficult intubation who underwent fiberoptic intubation or videolaryngoscopy as initial airway management, were excluded. 

### Data Analysis

All parameters were included in an Excel database and the correlations were performed in Excel, by using linear correlation equations and calculating Pearson correlation coefficient. For the diagnostic accuracy comparison, we conducted Receiver Operating Curves (ROC) analysis to compare the predictive values of HMDR_clin_ versus HMDR_echo_ for the occurence of grades III and IV Cormack–Lehane using the on-line MedCalc analysis software (Medcalc^®^, Medcalc Software Ltd., Ostend, Belgium). The area under the ROC curve (AUC) reflects the probability of correctly predicting the outcome variable and is a global measure of the performance of a diagnostic tool and a comparative method between two diagnostic or prognostic measurements [3,22]. The unpaired t-test was used to establish differences between continuous variables and Chi-squared test was used to establish differences between incidences. *p*-values lower than 0.05 were considered significant. 

## 3. Results

A total number of 160 adult patients (aged 19–89 years, from which 93 females) were prospectively included. From these, 21 patients (13.12%) presented BMI higher than 40 kg/m^2^, thus considered as having morbid obesity. Patients with morbid obesity had significantly larger neck circumferences (Table 1).

Mallampati scores of III and IV were significantly more frequent in the morbidly obese patients compared to patients without morbid obesity (*p* = 0.016). A difficult view during laryngoscopy-grades III and IV Cromack–Lehane was also more frequent, but without reaching statistical significance (*p* = 0.60) (Table 1). All patients were intubated during the first attempt or the second attempt with the help of a videolaryngoscope in the difficult cases. None of the patients presented desaturation during intubation. All patients had optimal mouth opening. In one case, a non-obese patient with a Cormack–Lehane score of III also presented limited neck extension.

For the patients without morbid obesity, mean HMDR_clin_ was 1.40, while for the patients with morbid obesity mean HMDR_clin_ was 1.41, showing no significant differences (t-test, *p* = 0.92) (Table 1). Similarly, there were no differences for the HMD measured clinically in neutral position or maximal hyperextension between patients with and without morbid obesity.

For the ultrasound measurements, HMD measured in neutral position did not differ between the two groups, but HMD measured with the head extended did. HMDR_echo_ was 1.26 for the patients without morbid obesity, while for the morbidly obese patients HMDR_echo_ was significantly lower 1.20 (t-test, *p* = 0.00314) (Table 1). 

The correlation between clinical and ultrasound measurements of the HMDR in patients without morbid obesity is described by the linear equation y = 0.3673x + 0.7507, where x = HMDR_clin_ and y = HMDR_echo_, with the Pearson correlation coefficient of 0.494 (moderate, fair positive relationship) (Figure 2).

The correlation between clinical and ultrasound measurements of the HMDR in patients with morbid obesity is described by the linear equation y = 0.0323x + 1.1559, with a Pearson correlation coefficient of 0.14, suggesting a weak or negligible association (Figure 3).

From the comparative ROC curve analysis conducted to compare the accuracy of HMDR_clin_ versus HMDR_echo_, a slightly higher area under the curve (AUC) for HMDR_echo_ was observed, 0.64 [95%CI 0.56–0.71], compared to HMDR_clin_ AUC of 0.52 [95%CI, 0.44–0.60], without reaching statistical significance (*p* = 0.34) (Figure 4).

## 4. Discussion

HMD is one of the clinical parameters of interest during the pre-anesthetic evaluation and its use has the advantage of being easy to perform. During maximal head extension, the hyoid bone position moves parallel in relation to the cervical spine, thus the expansion of the submandibular space reflects the ability to perform neck hyperextension. HMDR reflects the occipito-atlantoaxial complex extension capacity [3]. HMDR has been used to estimate the size of the submandibular space [7]. From this point of view, this might seem to be static, but it is actually a dynamic parameter, as the submandibular space expands during laryngoscopy. The elasticity in the saggital plane might reflect the submandibular space compliance, as described by Greenland et al. [23].

### 4.1. Clinical Studies

The HMDR discriminative cutoff was first determined in a clinical study conducted by Huh et al., who identified an optimal threshold of 1.2 as providing the optimal accuracy—a sensitivity of 88% and specificity of 60% [3]. Subsequent clinical studies using the same cutoff identified large variations in terms of sensitivity and specificity, depending on populations included, even if the methodology was the same. Some studies demonstrated high sensitivity, which is of interest in difficult airway investigations as false negatives can lead to catastrophic results. Good sensitivity and specificity values were found by some authors 86.3–95.6% sensitivity and 69.2% specificity in non-obese patients [4,5]. Other studies confirmed moderate sensitivity of around 60% and lower specificity, suggesting that HMDR has little utility for difficult airway prediction [6,7]. A low sensitivity of 27.78% has also been reported for HMDR_clin_ [8].

### 4.2. Imaging Studies

Due to the wide variability of the clinical studies and the lack of a reliable clinical tests to predict difficult airways, imaging techniques might be of help. Computed tomography, magnetic resonance imaging, and plain radiography have all been investigated [24]. Ultrasound is comparable to these, but is a cheaper, faster, non-irradiating, and non-invasive technique [25]. The sonographic assessment of the airway has encouraging results in predicting difficult laryngoscopy [9]. HMDR obtained by ultrasonography can be used for difficult airway prediction [10]. Because of the wide variability in clinical studies, ultrasound evaluation of the HMDR might be of interest, especially that the scanning technique is simple. With a cutoff of 1.24, the sensitivity of ultrasound-measured HMDR was 86–100%, while the specificity was 72–90.5% in the non-obese and obese populations [10,11]. Using ROC analysis, Koundal et al. found a cutoff of 1.08–1.085 for HMDR_echo_, with 65–75% sensitivity and 77–85% specificity [9,12]. However, a low sensitivity of 42.9% has also been reported, leading to the conclusion that these individual sonographic parameters, among others, have unsatisfactory diagnostic profiles [13].

Even if the evaluation of the HMDR seems to be comprehensive, the variability of the results does not allow clinicians to draw a definitive conclusion regarding the usefulness of the HMDR use for difficult airway prediction in practice. Moreover, comparative studies between clinically and ultrasound measurements of HMDR are not yet available. We found moderate correlation between HMDR_clin_ versus HMDR_echo_ for patients without morbid obesity and weak correlation between these two investigated parameters for patients with morbid obesity. This negligible relationship between HMDR_clin_ and HMDR_echo_ in the morbidly obese patients might be due to the fact that palpation of the hyoid bone is imprecise in the obese. Moreover, in the patients without morbid obesity, there are still cases of difficulties in palpating the anatomical structures of the neck, which might explain the modest association between the two investigated parameters.

The overall incidence of difficult view on laryngoscopy varies around 5%. However, in obese patients, higher rates have been reported up to 15% [14,15,26]. Obese patients display a series of physiological alterations, including increased oxygen consumption, reduced compliance of the chest wall, decreased functional residual capacity, and thus a higher chance of hypoxemia during airway management [22]. In our patients, HMDR_clin_ was not different between patients with and without morbid obesity, while HMDR_echo_ was discriminative. Wojtaczek et al. first suggested that HMDR might be more important in the obese population in a study investigating 12 obese patients, as the hyoid bone is more difficult to palpate and HMDR_echo_ seems to be a good discriminator between patients with easy versus difficult laryngoscopy [18]. Thus, in this patient category, ultrasound derived measurements might be most important. In the obese patients, fat pads around the neck and deposited anterior to the trachea might lower submandibular space compliance and might limit optimal head extension. This might be the reason why several ultrasound parameters in the anterior neck region might be correlated with difficult airways. Among these, the soft tissue thickness measured anterior to the epyglottis, tracheal wall, vocal cords commisure, and hyoid bone have been investigated. This association was first described by Ezri et al. for the obese [27].

In our cohort, obese patients had 9.52% Cormack–Lehane grades of III and IV, while patients without morbid obesity presented an incidence of 6.47% (*p* = 0.60). Because the sample size included was modest and the incidence of difficult view during laryngoscopy is low, even if patients with morbid obesity seem to have higher rates of grades III and IV Cormack-Lehane, this did not reach statistical significance. The correlation between HMDR_echo_ and HMDR_clin_ seems to be higher in patients without morbid obesity versus morbidly obese patients, but this might have limited consequences for airway management in clinical practice. The clinical impacts of these findings need to be further investigated. Based on our ROC curve analysis, we found no statistical significance for the difference between the AUC for HMDR_echo_ versus HMDR_clin_, even if the AUC for HMDR_echo_ seems to be higher. This is due to the modest number of patients included, which do not confer a high power for our study. Still, these figures might serve a larger future study to approximate the number of patients that should be included to have optimal power. The results obtained from the ROC analysis highlight modest figures for the AUC for both HMDR_clin_ and HMDR_echo_. Ultrasonographic parameters have limited ability to predict difficult airways when used alone, as previously stated by Andruskievicz et al. [13]. Still, the investigation of ultrasonographic-derived parameters might lead to composite scores that include both clinical and imaging parameters, tools that might have better predictive accuracy for difficult airway. There are various definitions of difficult airway, but Cormack–Lehane scale has been the outcome measure in difficult airway studies. Difficulty in intubation is generally associated with a difficult view (namely Cormack–Lehane grades III and IV observed during standard laryngoscopy) [6]. Other limits of the clinical and ultrasound studies are that the diagnostic methods are affected by both intrarater and intersubject variability. Moreover, ultrasound techniques, especially, are considered user-dependent. The unavailability of specified scanning protocols is also an important variable [12]. We consider that these have to be taken into account when a test of wide utility and use is designed to be implemented in clinical practice. The conduction of large multicenter studies with several different investigators from different centers could overcome these shortcomings. A perspective of value would be to combine clinical scores with ultrasound parameters in order to anticipate difficult airways. 

## 5. Conclusions

The association between ultrasonographically measured hyomental distance ratio and clinical measurements is moderate in patients without morbid obesity, but negligible in morbidly obese patients. These might be explained by difficulties in palpating anatomical structures of the airway. 

## Figures and Tables

**Figure 1 diagnostics-10-00140-f001:**
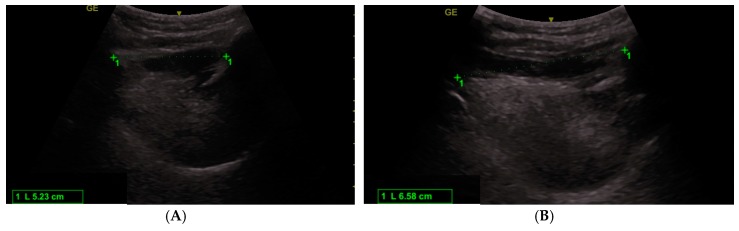
Ultrasound measurement of the hyomental distance in neutral position (**A**) and maximal hyperextended position (**B**).

**Figure 2 diagnostics-10-00140-f002:**
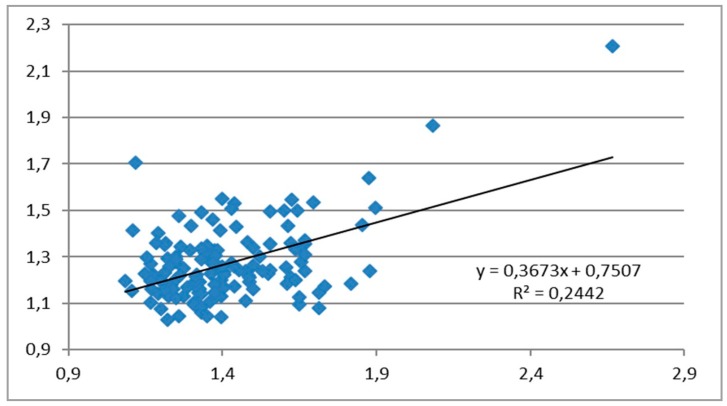
Correlation between the clinically measured hyomental distance ratio (HMDR_clin_) and ultrasound-measured hyomental distance ratio (HMDR_echo_) in patients without morbid obesity.

**Figure 3 diagnostics-10-00140-f003:**
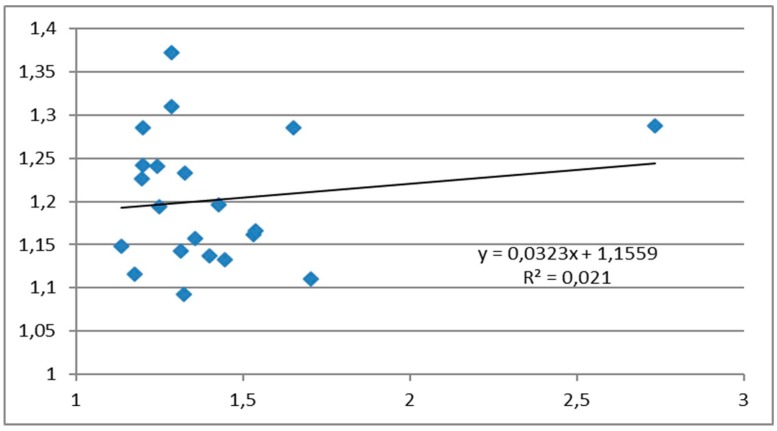
Correlation between HMDR_clin_ and HMDR_echo_ in patients with morbid obesity.

**Figure 4 diagnostics-10-00140-f004:**
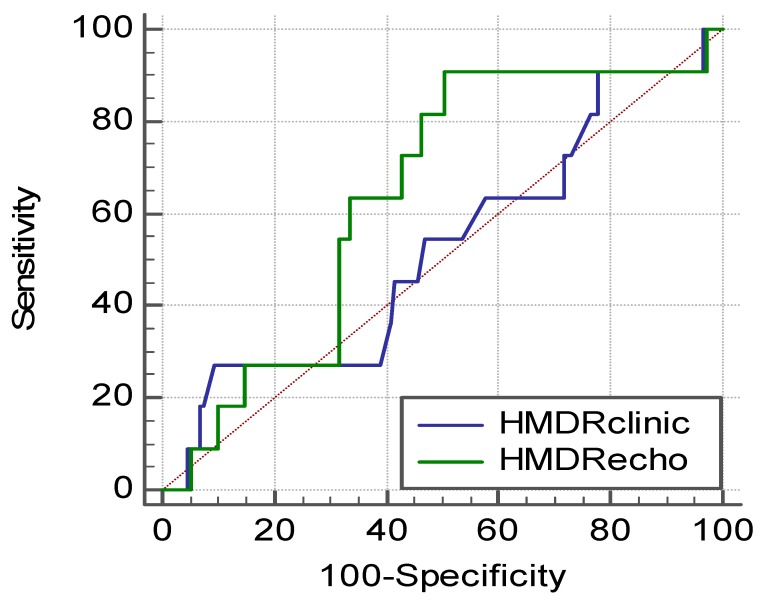
Receiver Operating Characteristics curves for HMDR_clin_ and HMDR_echo_ as discriminative variables and grades III and IV Cormack–Lehane as outcome variables.

**Table 1 diagnostics-10-00140-t001:** Comparative data for patients with and without morbid obesity. N: Number of patients included in each group; BMI: Body mass index; ULBT: Upper lip bite test; HMD: Hyomental distance; clin: Clinical measurement; echo: Ultrasound measurement; *p*-values derived from t-test for continuous data and Chi-squared test for frequencies.

	Patients without Morbid Obesity	Patients with Morbid Obesity	*p*-Value
N	139	21	
BMI (kg/m^2^)	28.23 ± 4.93	48.30 ± 7.45	<0.001
Neck circumference (cm)	42.56 ± 5.04	47.86 ± 5.11	<0.001
Mallampati scores 3 and 4	27/139 (19.42%)	9/20 (42.85%)	0.016
ULBT	14 edentolous patients 1/125	1 patient edentolous 0/20	0.54
HMD_clin_ neutral	4.36 ± 0.90	4.54 ± 1.04	0.46
HMD_clin_ extended	6.09 ± 1.31	6.34 ± 1.59	0.50
HMDR_clin_	1.40 ± 0.21	1.41 ± 0.33	0.92
HMD_echo_ neutral	4.16 ± 0.49	4.75 ± 0.47	0.54
HMD_echo_ extended	5.23 ± 0.59	5.71 ± 0.63	0.0036
HMDR_echo_	1.26 ± 0.16	1.20 ± 0.07	0.00314
Cormack–Lehane score III/IV	9/139 (6.47%)	2/21 (9.52%)	0.60

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
