# Peer review of "Clinical versus Ultrasound Measurements of Hyomental Distance Ratio for the Prediction of Difficult Airway in Patients with and without Morbid Obesity"

_diagnostics, 2020, doi:10.3390/diagnostics10030140_

Round 1

Reviewer 1 Report

In table 1 the authors showed hyomental distance and hyomental ratio with ultrasound is significantly different between two groups between without morbid obesity and without morbid obesity significant difference in between morbidly obese patients and not obese patients. They showed simply average and SD in both groups. And they showed Mallampati scores 3 and 4 were more frequent in patients with morbid obesity than patients without morbid obesity. But Cormack-Lehane score III/IV did not show significant different between two groups.

It is reasonable that ultrasound measurement of hyomental distance and hyomental ratiomore accurately than clinical measurement of them because of abundant fat tissue of theneck caused difficulty of palpation of hyoid bone

But no diagnostic capability of hyomental distance or hyomental distance with ultrasound to predict was not shown in the section of results, they described cut off level of 1.2 for the

prediction of difficulty of tracheal intubation according to their previous data. Also they didnot described cut off level of clinical (physical) measurement of hyomental distance and ratio

Posterior part of their conclusion “Ultrasound seems to be more precise than clinical palpation for hyomental distance ratio” cannot be guaranteed by their data

"imagistic" is inappropritae expresion, must be replaced by "imaging"

Author Response

Thank you very much for your work with analyzing our text. Please find below a point-by-point response:

  1. „In table 1 the authors showed hyomental distance and hyomental ratio with ultrasound is significantly different between two groups between without morbid obesity and without morbid obesity significant difference in between morbidly obese patients and not obese patients. They showed simply average and SD in both groups. And they showed Mallampati scores 3 and 4 were more frequent in patients with morbid obesity than patients without morbid obesity. But Cormack-Lehane score III/IV did not show significant different between two groups.”

It is true that, as expected, BMI and neck circumference differ significantly between the two groups. Mallampati also differs as scores. Hyomental distances and their ratios also differ- this is in accordance with others previous studies like that of Wojtaczek who first concluded that morbidly obese have different characterisics regarding HMD and HMDR and first described the ultrasound use to investigate HMDR in the obese. Regarding the difficult intubation/difficult laryngoscopy patients in obese versus non-obese, there are several reports in the literature, some authors arguing that difficult intubation is mpre freqent in the obese- 10%, compared to non-obese 5%. Other authors identify similar rates for Cormack. This is also our case here, but the results cannot be generalised as we only have a small sample size here of obese patients. In the discussion section we have added:

„In our cohort, obese patients had 9.52% Cormack-Lehane grades of III and IV, while patients without morbid obesity presented an incidence of 6.47% (p=0.60). Because the sample size included was modest and the incidence of difficult view during laryngoscopy is low, even if patients with morbid obesity seem to have higher rates of grades III and IV Cormack-Lehane, this did not reach statistical significance.”

  1. It is reasonable that ultrasound measurement of hyomental distance and hyomental ratiomore accurately than clinical measurement of them because of abundant fat tissue of theneck caused difficulty of palpation of hyoid bone

Thank you- this was also our hypothesis and we found the AUC to be higher for HMDRecho, even if we did not find a significant difference. As explained in the Results section, increasing the sample size – calculated based on our data here- this difference might be demonstrated. Sometimes it is difficult in medical studies to demonstrate something that might seem reasonable or logical, and this is one situation possibly.

  1. But no diagnostic capability of hyomental distance or hyomental distance with ultrasound to predict was not shown in the section of results, they described cut off level of 1.2 for the prediction of difficulty of tracheal intubation according to their previous data. Also they didnot described cut off level of clinical (physical) measurement of hyomental distance and ratio

The clinical cutoff of 1.2 was decribed by Huh et al. A close cutoff of 1.24 was identified for the ultrasound measurements and reported in the EJA. However, no previous study included both clinical and ultrasound measurements to compare. This is the novelty of our study here. We did not intend with this study to investigate again sensitivity, specificity – diagnostic accuracy- of these techniques of measurement. The aim was simply to assess differences between the two methods.

  1. Posterior part of their conclusion “Ultrasound seems to be more precise than clinical palpation for hyomental distance ratio” cannot be guaranteed by their data

We agree with your observation and we have deleted the last part of the conclusion section.

  1. "imagistic" is inappropritae expresion, must be replaced by "imaging"

We have changed this.

Thank you very much for your time allocated and the review performed, our intention is to improve our paper and if there are further questions or remarks, please do not hesitate to contact us.

Reviewer 2 Report

The authors described the usefulness of ultrasound measurements of hyomental distance ratio for the prediction of difficult airway in patients with and without morbid obesity. The reviewers also think measurements of hyomental distance ratio analysis was a usefulness method for the prediction of difficult airway. However, there are several problems in this study.

 Major comments

  1. In this study, the correlation between clinical and ultrasounds measurements in patients both with and without obesity were very low. I think that Pearson correlation coefficient of 0.494 and 0.14 were significant statistically, but not meaningful and not useful for real clinical site.
  2. And also, the result of ROC analysis was also not good results. AUC for HMDRecho (0.64) was not good. According to the figure 4, the sensitivity and specificity were about 60-70%.
  3. In this study, it is unclear what is primary outcome? Please describe precisely.
  4. In this study, it is unclear why patients considered to have difficult intubation who underwent fiberoptic intubation or videolaryngoscopy as initial airway management were excluded? These patients were real difficult airway.

Author Response

Thank you very much for your response and appreciation of the our work. Please find below a point-by-point detailed response, with changes highlighted in red:

  1. In this study, the correlation between clinical and ultrasounds measurements in patients both with and without obesity were very low. I think that Pearson correlation coefficient of 0.494 and 0.14 were significant statistically, but not meaningful and not useful for real clinical site.

We agree with your observation- the results are significant from the mathematical point of view, but their use in everyday practice might not have a significant impact on the clincial management of the patients. It highlights, though differences between obese and non-obese. Still, these difference might have low impact on the clincial management. We have now explained this in the Discussion:

„Even if the correlation between HMDRecho and HMDRclin seems to be higher in patients without morbid obesity versus morbidly obese patients, the usefulness of this finding might have limited consequences for airway management in clinical practice. The clinical impact of these findings need to be further investigated.  ”

  1. And also, the result of ROC analysis was also not good results. AUC for HMDRecho (0.64) was not good. According to the figure 4, the sensitivity and specificity were about 60-70%.

Indeed, the results are not great and the AUC are not good- for difficult airway prediction it is known that no single predicitive parameter (either clinical or sonographical) has optimal accuracy. For the ultraosund measurements, several other authors have reported modest results, includin Andruskievicz (reference 13). This is why it is mentioned frequently in papers on airway assesssment that the combined use of clinical and paraclincial parameters might be of help, or combined scores.

We have added now in the Discussion:

„The results obtained from the ROC analysis highlight modest figures for the AUC for both HMDRclin and HMDRecho. Ultrasonographic parameters have limited ability to predict difficult airway when used alone, as previously stated by Andruskievicz et al [13]. Still, the investigation of ultrasonographic-derived parameters might lead to composite scores that include both clinical and imaging parameters, tools that might have better predictive accuracy.”

  1. In this study, it is unclear what is primary outcome? Please describe precisely.

We have described the aims of this study in the last part of the introduction as

“The aim of our study is to assess and describe the correlation between HMDRclin and HMDRecho in patients with and without morbid obesity and to compare their diagnostic accuracy for the prediction of difficult airway.”

In our opinion, this is a comparative study fro two parameters in two different populations. It is not a diagnostic study trying to replicate previous clinical or ultrasound studies, but to compare the two- this is the novelty of this work. The two parameters clinical versus ultrasound, have not been compared in previous studies.

  1. In this study, it is unclear why patients considered to have difficult intubation who underwent fiberoptic intubation or videolaryngoscopy as initial airway management were excluded? These patients were real difficult airway.

We only included those patients in whom the attending anesthesiologist intended the direct laryngoscopy technique fromm the beginning. Those in whom the intial attempt with videolaryngoscopy or fiberoptic intuabtion were decided by the attending anesthesiologist were excluded, as we have explained in the Mehtods: “Patients considered to have difficult intubation who underwent fiberoptic intubation or videolaryngoscopy as initial airway management, were excluded. “

Thank you for the revision of our manuscript and all the time and work with our paper. Should there be any other remarks, please do not hesitate to contact us. We are aware that the quality of our manuscript needed improvement and we thank you for the remarks.

Round 2

Reviewer 1 Report

I am stisfied with correction that the authors made

Reviewer 2 Report

The revised manuscripts are well improved enough to publish.